# Nucleosomes impede Cas9 access to DNA in vivo and in vitro

Max A Horlbeck[1,2,3,4†], Lea B Witkowsky[5,6,7†], Benjamin Guglielmi[5,6,7], Joseph M Replogle[1,2,3,4], Luke A Gilbert[1,2,3,4], Jacqueline E Villalta[1,2,3,4], Sharon E Torigoe[5,6,7], Robert Tjian[5,6,7*], Jonathan S Weissman[1,2,3,4*]

[1]Department of Cellular and Molecular Pharmacology, University of California, San Francisco, San Francisco, United States; [2]California Institute for Quantitative Biomedical Research, University of California, San Francisco, San Francisco, United States; [3]Howard Hughes Medical Institute, University of California, San Francisco, San Francisco, United States; [4]Center for RNA Systems Biology, University of California, San Francisco, San Francisco, United States; [5]Howard Hughes Medical Institute, University of California, Berkeley, Berkeley, United States; [6]Department of Molecular and Cell Biology, University of California, Berkeley, Berkeley, United States; [7]CIRM Center of Excellence, University of California, Berkeley, Berkeley, United States

**Abstract** The prokaryotic CRISPR (clustered regularly interspaced palindromic repeats)-associated protein, Cas9, has been widely adopted as a tool for editing, imaging, and regulating eukaryotic genomes. However, our understanding of how to select single-guide RNAs (sgRNAs) that mediate efficient Cas9 activity is incomplete, as we lack insight into how chromatin impacts Cas9 targeting. To address this gap, we analyzed large-scale genetic screens performed in human cell lines using either nuclease-active or nuclease-dead Cas9 (dCas9). We observed that highly active sgRNAs for Cas9 and dCas9 were found almost exclusively in regions of low nucleosome occupancy. In vitro experiments demonstrated that nucleosomes in fact directly impede Cas9 binding and cleavage, while chromatin remodeling can restore Cas9 access. Our results reveal a critical role of eukaryotic chromatin in dictating the targeting specificity of this transplanted bacterial enzyme, and provide rules for selecting Cas9 target sites distinct from and complementary to those based on sequence properties.

*For correspondence: jmlim@
berkeley.edu (RT); Jonathan.
Weissman@ucsf.edu (JSW)

†These authors contributed
equally to this work

Competing interest: See
page 17

Reviewing editor: Karen
Adelman, National Institute of
Environmental Health Sciences,
United States

## Introduction

CRISPR (clustered regularly interspaced palindromic repeats) prokaryotic adaptive immune systems have yielded transformative tools for manipulating eukaryotic genomes. Most notably, the CRISPR-associated Cas9 protein from *Streptococcus pyogenes*, together with a single chimeric guide RNA (sgRNA), provides a programmable endonuclease that has revolutionized our ability to edit genomes (*Doudna and Charpentier, 2014*). Cas9 has been further modified to a nuclease-dead form (dCas9) to provide a programmable DNA-binding protein that can be fused to effector domains, making it possible to turn on or off targeted genes, mark specific genomic loci with fluorescent proteins, or alter epigenetic marks (*Doudna and Charpentier, 2014*; *Qi et al., 2013*; *Gilbert et al., 2013*; *Maeder et al., 2013*; *Chen et al., 2013*; *Ma et al., 2015*; *Hilton et al., 2015*; *Kearns et al., 2015*). A central challenge in implementing these tools is identifying effective and specific sgRNAs. While much of the effort to define relevant rules has focused on the sequence of the target site and sgRNA

**eLife digest** Many bacteria have a type of immune system known as CRISPR that can target and cut foreign DNA to protect it against viruses. Recently, the CRISPR system was adapted to allow scientists to easily manipulate the genome of humans and many other organisms. However, unlike the loosely organized DNA found in bacteria, the DNA that makes up the human genome is tightly packed and wrapped around complexes of proteins to form structures called nucleosomes. It was not clear whether the CRISPR system was able to effectively target the stretches of DNA in a nucleosome.

In 2013, researchers developed a modified version of CRISPR, known as CRISPR interference, to block gene activity and in 2014 used it to systematically repress many of the genes in the human genome. Now, Horlbeck, Witkowsky et al. – who include several of the researchers from the 2014 work – have analyzed existing data for a specific type of human cell grown in the laboratory and found that CRISPR interference activity was strongest in certain areas around the start of each gene. However, CRISPR interference was much weaker in other areas of genes that coincided well with stretches of DNA that are known to often be bound by nucleosomes. Nucleosomes also appeared to block CRISPR editing, although the effects were less pronounced.

Horlbeck, Witkowsky et al. then directly tested whether nucleosomes could prevent the CRISPR system from binding or modifying the DNA. When the individual components were mixed in test tubes, the CRISPR system could readily target "naked" DNA. However, it could not access nucleosome-bound DNA, unless an enzyme that can move nucleosomes along the DNA in the human genome was also added to the mix. These findings suggest one way that CRISPR can manipulate much of the human genome despite the widespread presence of nucleosomes. Future work will now aim to develop computational methods that take the positions of nucleosomes into account when picking DNA sites to target with CRISPR.

(*Chari et al., 2015*; *Doench et al., 2014*; *Wang et al., 2014*; *Xu et al., 2015*), these only partially predict Cas9 activity and suggest that additional determinants likely exist.

Chromatin structure may represent a key parameter governing Cas9 efficacy in eukaryotic cells. CRISPR evolved in archea and bacteria (*Makarova et al., 2015*) and is likely not optimized to explore and modify large, chromatin-bound eukaryotic genomes, a hypothesis supported by several studies that point to a correlation between rates of DNA binding and cleavage in regions of open chromatin as measured by DNase I hypersensitivity (*Chari et al., 2015*; *Singh et al., 2015*; *Wu et al., 2014*). Additionally, recent single-molecule imaging studies have shown that dCas9 explores euchromatin more frequently than it does heterochromatin (*Knight et al., 2015*).

We hypothesized that nucleosomes, the basic unit of chromatin structure, are an important impediment to Cas9 recognition. Here, we addressed this hypothesis in vivo by leveraging large datasets collected from over 30 sgRNA tiling and genome-scale genetic screens (*Gilbert et al., 2014*). We found that regions of high nucleosome occupancy in vivo, as determined by MNase-seq (micrococcal nuclease sequencing) (*Johnson et al., 2006*; *Valouev et al., 2011*), were strongly depleted of highly active sgRNAs for CRISPR interference (CRISPRi) (*Gilbert et al., 2013*; *2014*) and nuclease-active Cas9. We complemented these results with in vitro experiments demonstrating that formation of a nucleosome provides a direct and profound block to dCas9 binding and Cas9 cleavage. Despite this strong barrier to Cas9 activity, we found that addition of a chromatin remodeling enzyme to chromatinized DNA in vitro can restore Cas9 access to nucleosomal DNA, highlighting one route by which CRISPR may still be able to modify chromatin in vivo. Our results reveal a fundamental aspect of the mechanism by which this transplanted bacterial enzyme interacts with eukaryotic chromatin, and provide a new dimension for selecting highly active sgRNAs.

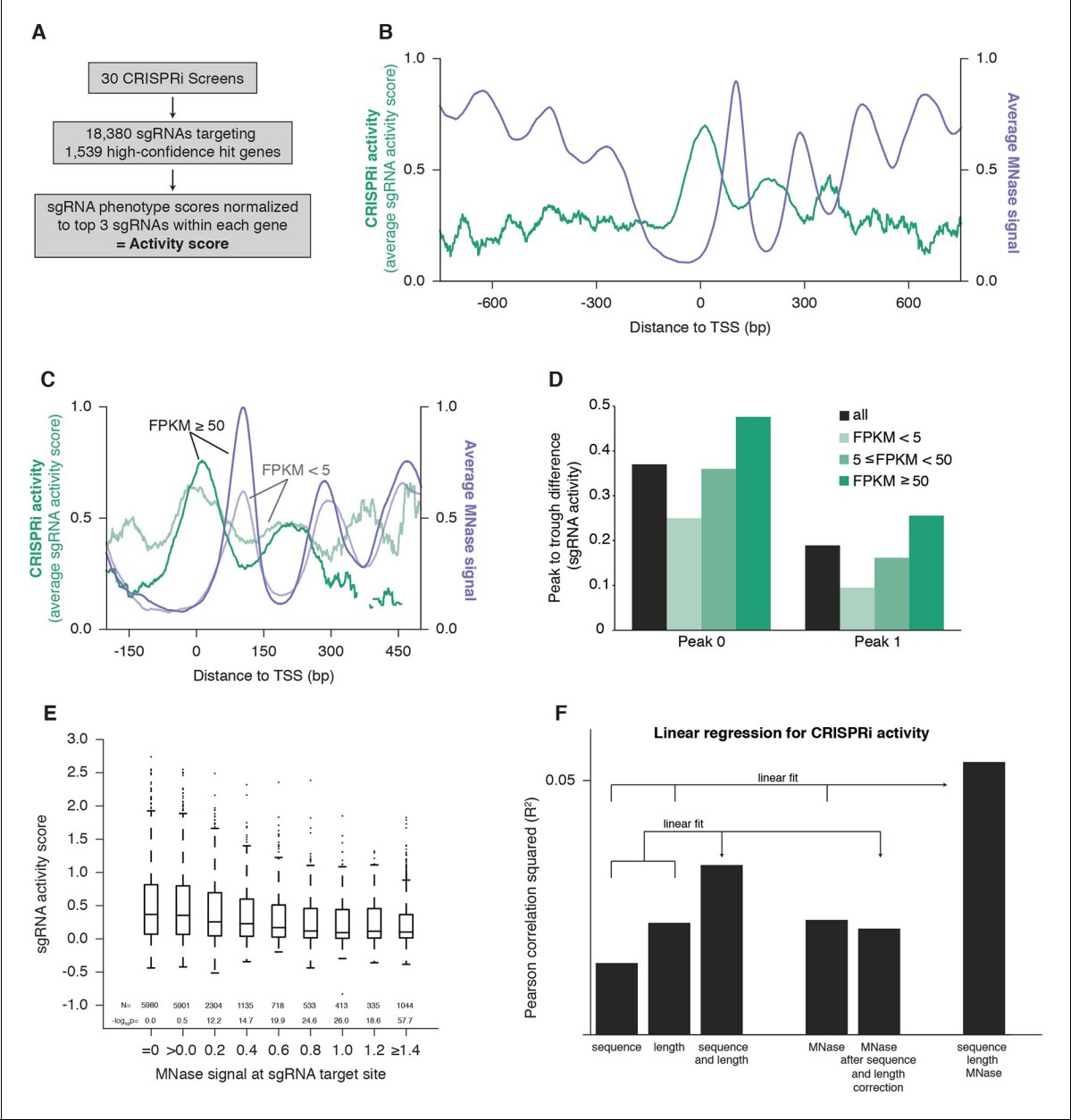

**Figure 1.** CRISPRi activity anti-correlates with nucleosome occupancy. (**A**) Workflow for generating CRISPRi activity scores from pooled genetic screens. The resulting values are distributed around 0 for inactive sgRNAs and around 1 for highly active sgRNAs. (**B**) Average CRISPRi activity and MNase-seq signal relative to the TSS. Green line represents average CRISPRi activity score of all sgRNAs within a 50 bp window around each position. Purple line represents the K562 MNase-seq signal at each position averaged across all genes analyzed. (**C**) Average CRISPRi activity and MNase-seq signal for genes grouped by expression value. Genes were grouped into lower expression (light lines; N=240), higher expression (heavy lines; N=368), and medium expression (omitted for clarity; N=930), and analyzed as in (**B**). Expression values were obtained as fragments per kilobase million (FPKM) from ENCODE K562 RNA-seq data. Average activity at positions with fewer than 10 sgRNAs within the 50bp window was not calculated. (**D**) Quantification of the amplitude of periodic CRISPRi activity. Peak and trough coordinates were obtained by calculating the local maxima and minima of the activity traces from analyses in (**B**) and (**C**). Peak 0 was defined as the local maximum closest to the TSS, peak 1 was defined as the next maximum downstream of peak 0, and troughs were defined as the minima immediately downstream of the respective peaks. (**E**) CRISPRi activity and target site nucleosome occupancy for individual sgRNAs. Target site nucleosome occupancy was calculated from the average MNase-seq signal at all genomic coordinates across the length of the sgRNA protospacer and the protospacer adjacent motif (PAM). sgRNAs were then binned by the target site nucleosome occupancy, displayed as box-and-whisker plots, and labeled according to the minimum value within the bin except where indicated. P-values were calculated by a two-tailed Mann-Whitney test comparing each bin to the =0.0 bin. (**F**) Linear regression for CRISPRi activity. The squared Pearson correlation was calculated for the sgRNA activity scores compared to the indicated individual parameters (bars 1, 2, and 4) or linear fits of multiple

*Figure 1 continued on next page*

*Figure 1 continued*

parameters (bars 3 and 6). sgRNA activity scores were corrected for sequence and length features (bar 5) by subtracting the linear fit of those two features.

## Results

### CRISPRi activity is periodic and out-of-phase with nucleosome positioning

In order to study the features of chromatin that affect CRISPRi activity, we integrated data from whole-genome screens testing for a wide range of phenotypes performed with a previously described CRISPRi library targeting each gene with ~10 sgRNAs (*Gilbert et al., 2014*). We selected 30 screens performed in the cell line K562 expressing dCas9-KRAB (ML, BA, BB, CYP, MK, YC, JF, and JN, personal communication) and set a threshold for high-confidence hit genes, which allowed us to assess the relative strength of phenotypes for the sgRNAs targeting these genes. Specifically, we analyzed 18,380 sgRNAs targeting 1539 genes, and generated 'activity scores' by normalizing sgRNA phenotypes to the average of the 3 sgRNAs with the strongest phenotypes for each gene (*Figure 1A* and *Supplementary file 1*). To assess how CRISPRi activity varies with respect to the transcription start site (TSS), we plotted the average sgRNA activity score as a function of distance to the FANTOM-annotated TSS (*Forrest et al., 2014*) (*Figure 1B*). This analysis revealed a robust, periodic pattern of activity, with peaks at ~190 bp intervals relative to the TSS. This periodicity was highly reminiscent of patterns previously described for nucleosomes (*Jiang and Pugh, 2009*). Indeed, analysis of K562 MNase-seq data from the ENCODE consortium (*Valouev et al., 2011*; *ENCODE Project Consortium, 2012*) revealed that the average nucleosome signal was strongly anti-correlated with CRISPRi activity (*Figure 1B*), suggesting that high nucleosome occupancy leads to low CRISPRi activity.

To further explore this inverse relationship between CRISPRi activity and nucleosome organization, we exploited the previous observation that nucleosome phasing is more pronounced in highly expressed genes (*Valouev et al., 2011*). We grouped genes by their expression in K562 (*ENCODE Project Consortium, 2012*; *Djebali et al., 2012*) and analyzed CRISPRi activity and nucleosome occupancy within each group. In support of a connection, we found that peak to trough amplitudes of both features were larger for highly expressed genes (*Figures 1C,D*).

We also analyzed the MNase signal at each sgRNA target site to determine whether nucleosome occupancy could explain variation in CRISPRi activity between individual sgRNAs (*Figure 1E*). Consistent with the hypothesis that nucleosomes exclude dCas9, almost all of the highly active sgRNAs targeted sites with low MNase-seq signal. However, not all sgRNAs targeting sites with low nucleosome occupancy were highly active, matching previous findings that sgRNA and target DNA sequence features also influence efficiency of the CRISPR system (*Chari et al., 2015*; *Doench et al., 2014*; *Wang et al., 2014*; *Xu et al., 2015*). To exclude the possibility that differences in sequence features alone between nucleosome-bound and -free regions could explain the periodicity of CRISPRi activity (rather than the presence of nucleosomes per se), we performed linear regression using MNase signal, sgRNA length (*Xu et al., 2015*; *Gilbert et al., 2014*), and a validated sgRNA sequence scoring algorithm (*Doench et al., 2014*). We found that each parameter individually correlated with sgRNA activity ($p < 10^{-58}$ for each; *Figure 1F*). Importantly, correction for sequence and length features had minimal impact on the ability of the MNase signal to predict CRISPRi activity ($p < 10^{-85}$ after correction). Indeed, a linear fit of all three features provided a still stronger correlation ($p < 10^{-221}$; *Figure 1F*, far right column), suggesting that incorporating nucleosome occupancy in future sgRNA design algorithms could significantly improve predictive value. We have recently developed a comprehensive algorithm for predicting highly active CRISPRi sgRNAs that, by accounting for nucleosome occupancy, higher order sequence features, and non-linear relationships in these parameters, shows even greater correlation with this dataset that was already enriched for active sgRNAs using our original CRISPRi library design principles (*Gilbert et al., 2014*) (cross-validation $R^2 = 0.32$; Horlbeck et al., manuscript in preparation).

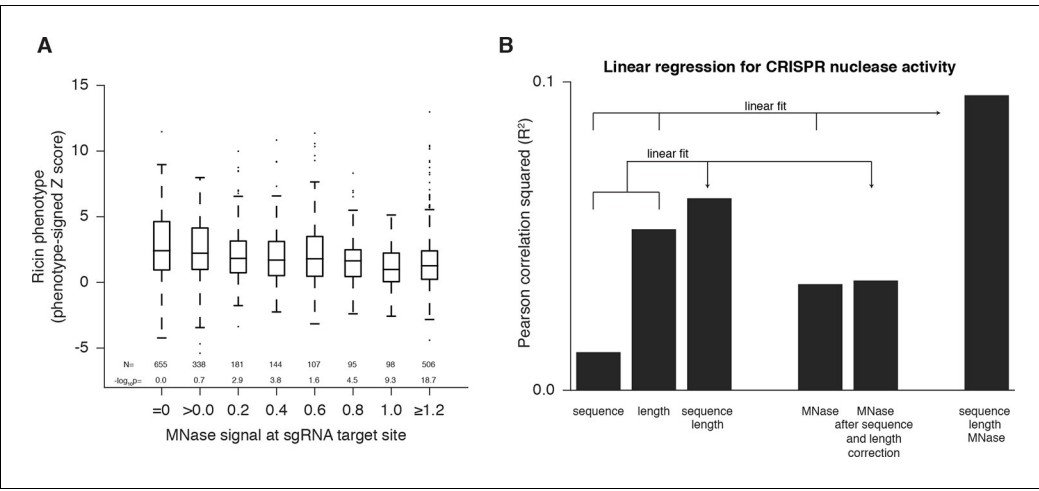

**Figure 2.** Cas9 nuclease activity anti-correlates with nucleosome occupancy. (**A**) Cas9 nuclease phenotypes and target site nucleosome occupancy for individual sgRNAs targeting CDS regions. Ricin susceptibility phenotypes for each sgRNA are expressed as a Z score and are positive if the phenotype matches the expected knockdown phenotype. Target site nucleosome occupancy was calculated as in *Figure 1E*. sgRNAs were then binned by the target site nucleosome occupancy, displayed as box-and-whisker plots, and labeled according to the minimum value within the bin except where indicated. P-values were calculated by a two-tailed Mann-Whitney test comparing each bin to the =0.0 bin. (**B**) Linear regression for Cas9 nuclease phenotypes. The squared Pearson correlation was calculated for the sgRNA activity scores compared to the indicated individual parameters (bars 1, 2, and 4) or linear fits of multiple parameters (bars 3 and 6). sgRNA activity scores were corrected for sequence and length features (bar 5) by subtracting the linear fit of those two features.

The following figure supplement is available for figure 2:

**Figure supplement 1.** Cas9 nuclease activity anti-correlates with nucleosome occupancy at all target sites.

## Nuclease-active Cas9 activity anti-correlates with nucleosome occupancy

In order to generate a dataset for evaluating the effect of nucleosome positioning on nuclease-active Cas9 in K562 cells, we took advantage of our previously described library densely tiling sgRNAs in 10kb windows around the TSS of 49 genes known to modulate susceptibility to the toxin ricin (*Gilbert et al., 2014*; *Bassik et al., 2013*) and tested Cas9-expressing cells for resistance or sensitivity to ricin (*Supplementary file 2*). We observed phenotypes consistent with the expected knockdown phenotype primarily in coding sequences (CDS) but also in some promoter regions, consistent with recent results showing that modifications introduced by Cas9 can disrupt *cis*-regulatory regions (*Canver et al., 2015*). Analysis of CDS-targeting sgRNAs revealed that strong phenotypes were found predominantly in regions of low MNase-seq signal (*Figure 2A*), although this relationship was less pronounced than for CRISPRi. This may be due to decreased phasing of nucleosomes within the gene body (*Jiang and Pugh, 2009*). When we analyzed all sgRNAs in the library, thus incorporating information from *cis*-regulatory regions where nucleosomes are well phased, we found the effect of nucleosome position to be even stronger (*Figure 2—figure supplement 1*). Although nucleosome occupancy was predictive of CRISPR activity independent of sgRNA sequence features, a much stronger correlation was obtained when all features were considered (*Figure 2B*). Therefore, nucleosome organization likely represents an important feature for CRISPR sgRNA design and should be considered a key contributing factor in interpreting future tiling mutagenesis experiments of coding and non-coding regions.

## Nucleosomes are sufficient to fully block cleavage by Cas9 in vitro

Our in vivo experiments reveal a strong anti-correlation between Cas9-mediated downstream phenotypic outputs and nucleosome positioning, but do not directly report on the ability of Cas9 to access nucleosomal DNA. Factors other than Cas9 access could contribute to the observed

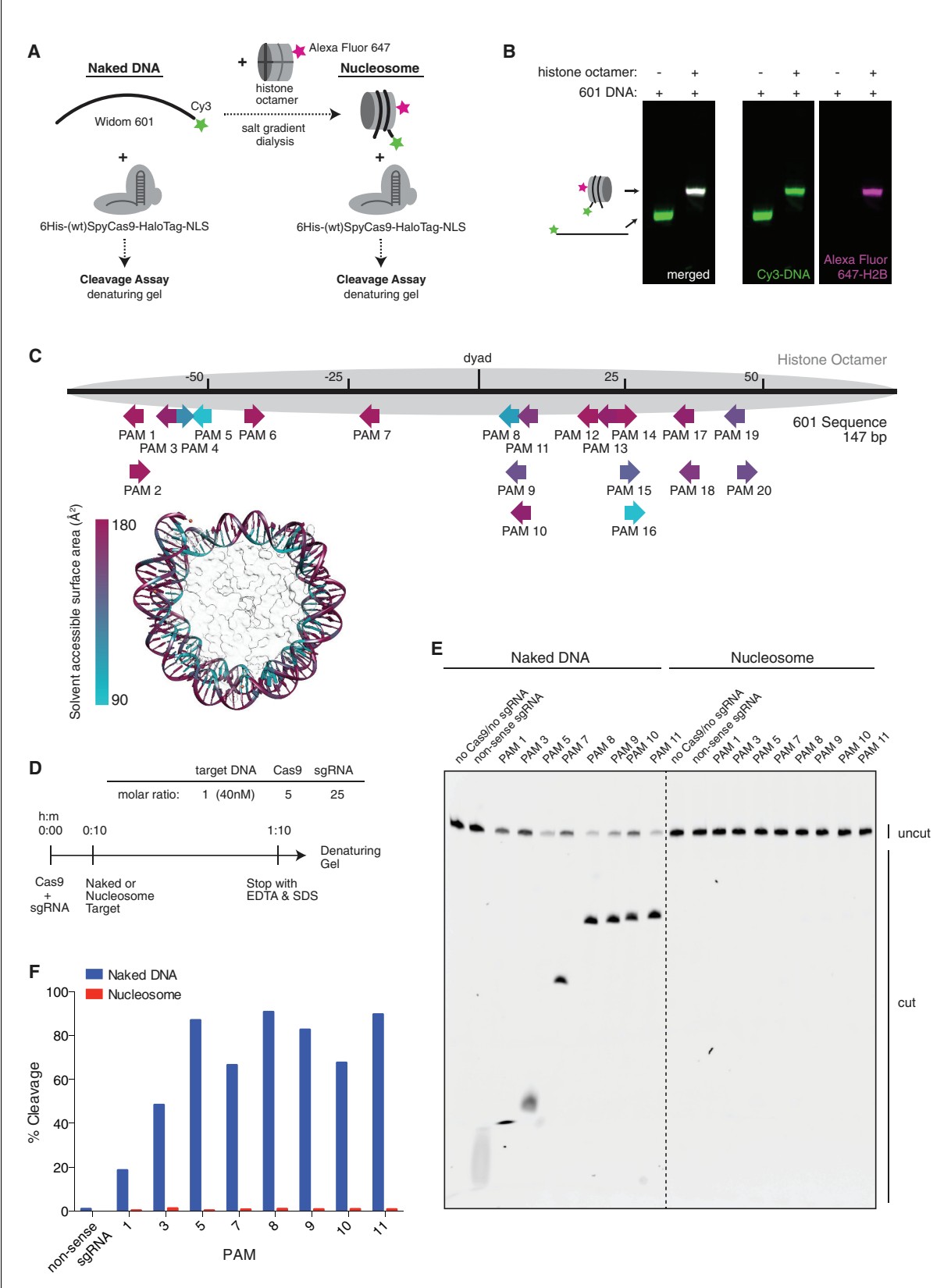

**Figure 3.** Cas9 nuclease activity is blocked by the presence of a nucleosome in vitro. (**A**) Schematic of the experimental setup for in vitro cleavage assays. Mononucleosomes were assembled by salt gradient dialysis of purified mouse histone octamer with the minimal nucleosome positioning

*Figure 3 continued on next page*

*Figure 3 continued*

sequence, Widom 601 (147 bp). Prior to assembly, DNA was 5'-end-labeled with Cy3 and histone H2B was fluorescently labeled using an introduced cysteine (T115C) coupled to Alexa Fluor 647. Purified His-tagged and HaloTagged Cas9 pre-loaded with in vitro transcribed sgRNAs were added to naked DNA or assembled nucleosomes, and DNA cleavage products were visualized using a denaturing gel imaged for Cy3-DNA fluorescence. See also *Figure 3—figure supplement 1A*. (B) Confirmation that fully occupied, well-positioned nucleosomes were assembled. After assembly using salt gradient dialysis, the produced nucleosomes were visualized using a native PAGE gel imaged in the Cy3 and Alexa Fluor 647 channels. Full incorporation of the free DNA into a nucleosome occupying a single position on the DNA is indicated by the presence of a single-shifted band containing all of the detectable Cy3-DNA and Alexa Fluor 647-H2B signal. (C) Available PAMs and solvent accessibility of the 601 nucleosome positioning sequence. (Above) A schematic of the 601 sequence. The location of the histone octamer when assembled into a nucleosome is indicated by the gray oval. The location of PAMs within the double-stranded sequence are indicated with arrows spanning the 3 bp of the PAM, pointing in the 5' to 3' orientation of the NGG motif. The arrows are colored according to solvent accessibility at the center of the PAM as calculated from the crystal structure of the 601 nucleosome (PDBID 3LZ0, *Vasudevan et al., 2010*). (Below) Crystal structure of the 601 nucleosome. For clarity, the surface area of the histones in the crystal structure has been made transparent. The DNA in the crystal structure is colored according to solvent accessibility using the same color scale as the PAMs above. Residues colored teal are less accessible, while residues colored fuchsia are more accessible by solvent. (D) Experimental conditions and timeline for cleavage assays. (E) Denaturing PAGE gel showing the results of a cleavage assay targeting the indicated PAMs. Cleavage reactions containing naked DNA were loaded on the left half of the gel, while reactions containing nucleosomes were loaded on the right. The DNA was imaged via a Cy3 fluorophore attached to the 5' end of the sgRNA-complimentary strand. A negative control was conducted with an sgRNA that had no sequence complementarity to the 601 sequence used (non-sense guide). See also *Figure 3—figure supplement 1B,C* for additional controls. (F) Quantification of the gel in (D). For each lane, percent cleavage was determined by calculating the percent of the total band signal corresponding to cleaved DNA.

The following figure supplement is available for figure 3:

**Figure supplement 1.** HaloTagged Cas9 activity is indistinguishable from untagged Cas9.

correlation in our data. For example, (d)Cas9 may bind or cut equally well in nucleosome-bound and un-bound regions, but may exert the observed modulation of gene expression through interference with transcriptional pausing, splicing, regulatory looping, or binding of important regulatory factors, processes which also correlate with nucleosome organization (*Jonkers and Lis, 2015*; *Denisov et al., 1997*; *Naftelberg et al., 2015*; *Segal and Widom, 2009*; *Tilgner and Guigó, 2010*; *Thurman et al., 2012*; *Hsieh et al., 2015*).

To determine whether (d)Cas9 activity is indeed directly affected by the presence of nucleosomes, we turned to a purified in vitro reconstituted system. Mouse histones were recombinantly expressed, purified, and assembled into a nucleosome using 147bp of the Widom 601 positioning sequence (*Figures 3A,B*) (*Lowary and Widom, 1998*). Conveniently, the 601 sequence contains numerous NGG protospacer adjacent motifs (PAMs) required for Cas9 recognition, spanning the full length of the DNA at different helical positions, allowing us to test the effects of position and solvent accesibility within the nucleosome (*Figure 3C*). We first tested the ability of Cas9 to cleave nucleosomal DNA. We pre-loaded purified Cas9 (histidine tagged and HaloTagged) with in vitro transcribed sgRNA, then introduced either naked 601 DNA or that same DNA assembled into a nucleosome (*Figures 3A,D* and *Figure 3—figure supplement 1A*). Fluorescent labeling of the DNA allowed us to visualize the cleavage products on a denaturing Urea-PAGE gel. In agreement with our in vivo data, the nucleosome protected its DNA from cleavage by Cas9, and complete protection from cleavage was observed to be independent of target position within the nucleosome (*Figures 3E,F* and *Figure 3—figure supplement 1B,C*). While our manuscript was under review, a study by Hinz and colleagues reported a similar in vitro finding that Cas9 nuclease activity is inhibited within the nucleosome but not at adjacent linker sequences (*Hinz et al., 2015*). Additionally, previous single-molecule and biochemical studies have established that nucleosomal DNA undergoes transient unwrapping or breathing at the entry and exit sites, creating a gradient of accessibility along the nucleosome (*Li and Widom, 2004*; *Li et al., 2005*; *Luger and Hansen, 2005*; *Choy and Lee, 2012*; *Tomschik et al., 2009*; *Polach and Widom, 1995*). This property is often credited for the observed position-dependent binding patterns of many transcription factors and DNA-binding proteins. Interestingly, our data suggests that the cleavage activity of Cas9 in vitro is not detectably influenced by this effect, although less stable nucleosomes than the Widom 601 nucleosome or target sites closer to the nucleosome edge may exhibit more breathing and thus more accessibility than those tested here.

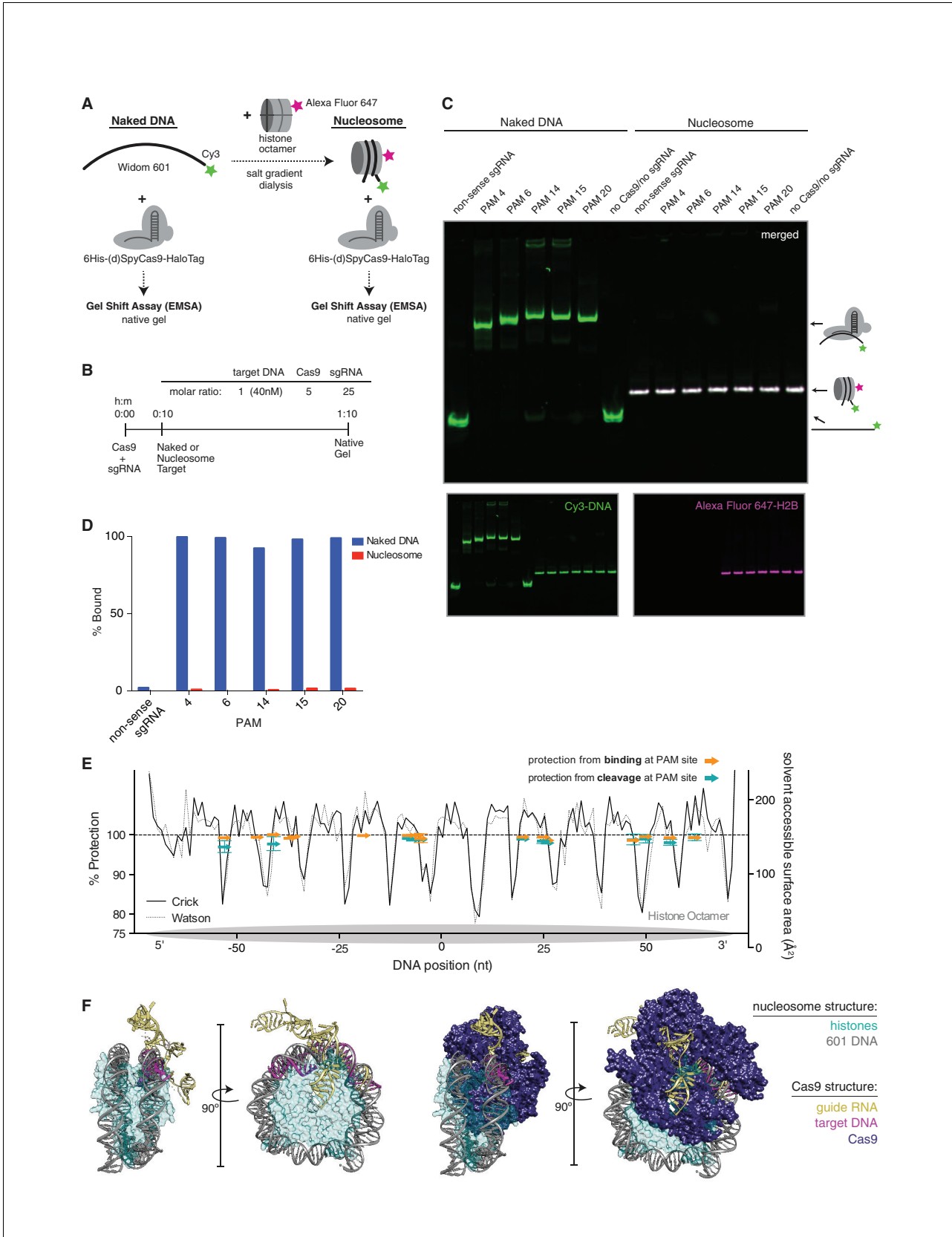

**Figure 4.** dCas9 is unable to bind nucleosomal DNA in vitro. (**A**) Schematic of the experimental setup for in vitro binding assays. Either naked DNA or assembled nucleosomes were incubated with catalytically dead Cas9 (dCas9, histidine tagged and HaloTagged), and binding was assessed by an

*Figure 4 continued on next page*

*Figure 4 continued*

electrophoretic mobility shift assay (EMSA). (B) Experimental conditions and timeline for binding assays. (C) A native PAGE gel showing the results of an EMSA in which dCas9 was targeted to the indicated PAMs on either naked or nucleosomal DNA. Gels were scanned for fluorescence from Cy3 on the DNA (green) and Alexa Fluor 647 on histone H2B (magenta). The two color channels were merged to identify the location of intact nucleosomes (white). See also *Figure 4—figure supplement 1* and *3* for reagent preparation and experimental conditions, and *Figure 4—figure supplement 2* for comparison with wtCas9 binding. (D) Quantification of the gel in (C). Percent bound was determined by calculating the percent of the total band signal in each lane corresponding to Cas9-bound target as determined by a shift in mobility within the gel. (E) Summary of binding and cleavage results for each PAM tested. The ability of Cas9 to bind or cleave nucleosomal DNA at a targeted PAM is displayed as percent protection by the nucleosome, and was calculated by taking the ratio of binding or cleavage on nucleosomal DNA to that on naked DNA. While only the largest error bars are visible, replicates were performed for 15 of the 30 data points and are displayed with error bars showing standard deviation from the mean. The DNA positions plotted correspond to the three nucleotides of the targeted PAM. In order to compare percent protection from binding and cleavage with solvent accessibility, the PAMs are overlaid with a plot of the solvent accessible surface area for each strand (Watson or Crick) of DNA in the 601 nucleosome structure. The percent protection at each PAM, as well as the solvent accessibility were plotted so that the 5' end of each DNA strand begins at the left of the graph, where position 0 indicates the dyad. (F) Structural assessment of the ability of Cas9 to bind nucleosomal DNA. Superposition of the Cas9-guideRNA-DNA crystal structure (PDBID 4UN3, *Anders et al., 2014*) onto the 601 nucleosome crystal structure (PDBID 3LZ0, *Vasudevan et al., 2010*) was achieved by alignment of the DNA path in both structures. (Left) To better view the alignment of the Cas9 target DNA with the nucleosomal DNA, the Cas9 protein density has been removed. (Right) After alignment of the DNA, inclusion of the Cas9 protein density reveals extensive steric clashes with the histones. Histone surface area was made partially transparent to better reveal the overlapping densities with Cas9.

The following figure supplements are available for figure 4:

**Figure supplement 1.** Quality controls.

**Figure supplement 2.** DNA binding by dCas9 is also representative of wtCas9 binding.

**Figure supplement 3.** (wt/d)Cas9 purification strategy.

## Cas9 is unable to bind nucleosomal DNA in vitro

Cleavage of DNA by Cas9 has been described as a stepwise process (*Wu et al., 2014*; *Knight et al., 2015*; *Sternberg et al., 2014*) in which Cas9 must first scan for PAMs, unzip the DNA duplex, and fully pair the guide RNA and target DNA prior to cleavage. While our in vitro data show that full pairing and cleavage is prevented by the presence of a nucleosome, we wondered if binding without cleavage, especially at the more accessible ends of the nucleosome, might still occur. Additionally, the first step in binding, PAM recognition, may be governed by helical location within the nucleosome as well as its proximity to the more dynamic ends. Histones make contacts with the DNA at every helical turn, thus a PAM may fall on the outside of the nucleosome, exposing it to solvent, or on the inside at the DNA-histone interface. To test the influence of target location within the nucleosome on Cas9 binding, we used an electrophoretic mobility shift assay to monitor binding of dCas9 to Cy3 end-labeled 601 DNA (*Figures 4A,B*), either free or assembled into a nucleosome containing Alexa Fluor 647-labeled histone H2B (*Figure 4—figure supplement 1A*). Consistent with our cleavage results, binding by dCas9 was abolished by the presence of the nucleosome, regardless of the targeted dCas9 binding site (*Figure 4C–E* and *Figure 4—figure supplement 2A,B*).

To better understand how a nucleosome might impede Cas9 binding, we aligned the available crystal structures of DNA-bound Cas9 and the structure of the 601 nucleosome (*Vasudevan et al., 2010*, PDB ID 3LZ0; *Anders et al., 2014*, PDB ID 4UN3). We superimposed the target DNA in the Cas9 crystal structure with the DNA in the nucleosome structure at a site where the two DNA paths gave the best fit. The resulting combined structure reveals that the Cas9 protein poses significant steric clashes with the histones (*Figure 4F*). Given the extent of overlapping densities in the two structures, it seems unlikely that the histones and Cas9 could co-occupy the same piece of DNA. Additionally, it may be important to note that unlike other DNA binders such as transcription factors, binding by Cas9 constitutes melting of target DNA, which may pose an additional barrier to binding on a nucleosome. This hypothesis leaves two possible outcomes of targeting Cas9 to nucleosomal DNA: either Cas9 is capable of displacing histones in order to engage nucleosomal DNA, or it is excluded altogether. Our data support the latter conclusion.

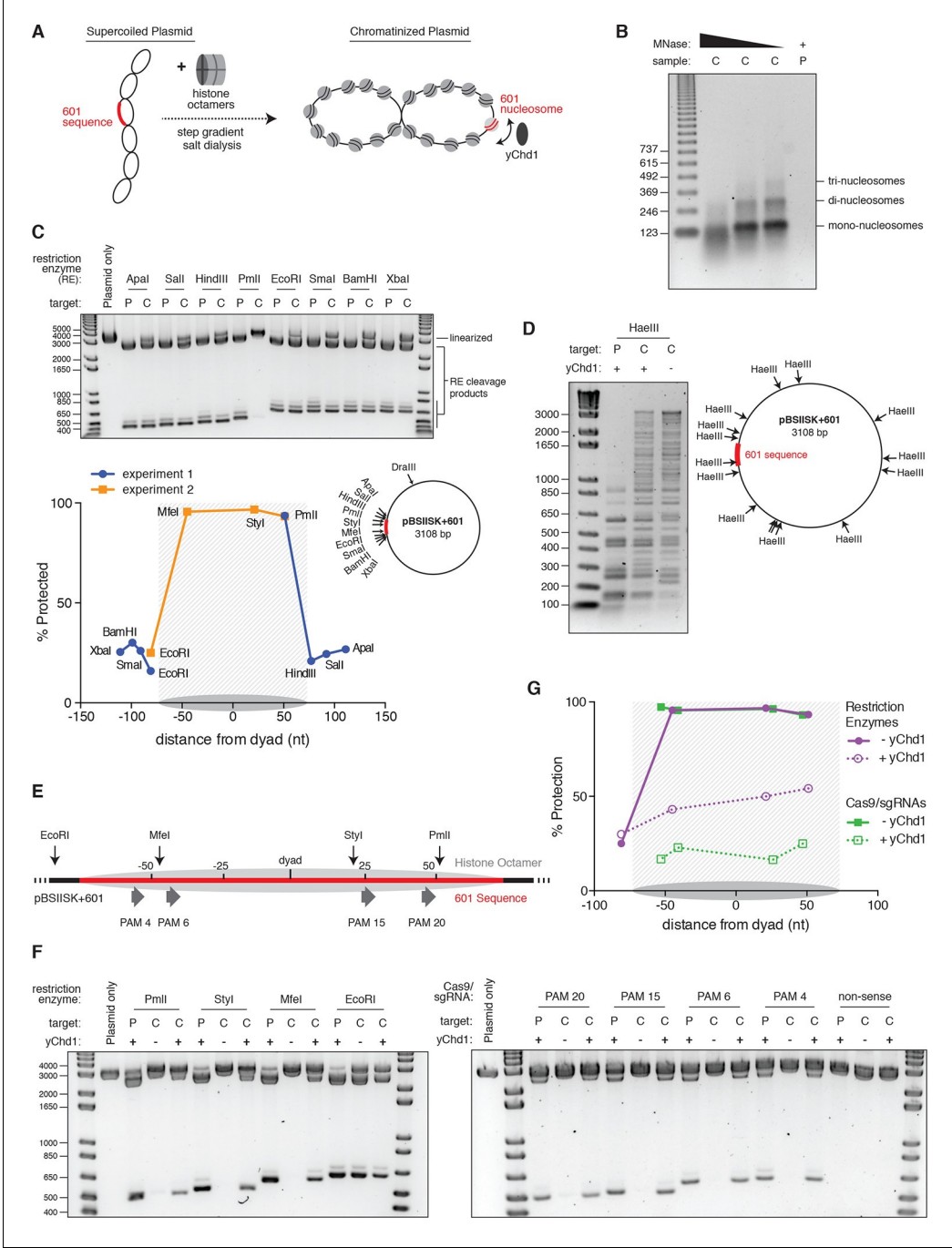

**Figure 5.** Nucleosomes within chromatinized DNA can block cleavage by Cas9, but a chromatin remodeling factor can restore Cas9 access. (**A**) Schematic of the experimental setup. Supercoiled plasmid containing the 601 sequence inserted into a pBlueScript II SK (+) backbone (pBSIISK+601) was chromatinized by step gradient salt dialysis in the presence of histone octamer. Purified yeast Chd1 (yChd1) remodeling factor was used to test the effect of ATP-dependent remodeling factors on Cas9 access to nucleosomal DNA. (**B**) Quality assessment of the chromatinized plasmid used in this study. Titrated amounts of Micrococcal Nuclease (MNase) were incubated with the chromatinized plasmid, and the resulting pattern of protection by assembled nucleosomes was visualized on a 1.3% agarose gel post-stained with ethidium bromide (EtBr). As a control, the supercoiled plasmid was also incubated with the lowest concentration of MNase. (**C**) A restriction enzyme accessibility assay (REAA) was used to assess the occupancy and position of the nucleosome assembled at the 601 sequence within the chromatinized plasmid. A panel of unique restriction enzyme sites spanning the 601 sequence were incubated with either the supercoiled plasmid, or the chromatinized plasmid. Cleavage was stopped, and protein was removed by

*Figure 5 continued on next page*

*Figure 5 continued*

incubation with proteinase K followed by Phenol:Chloroform:Isoamyl alcohol extraction and ethanol precipitation. (Top) The resulting DNA was then linearized using DraIII, and the level of cleavage by the restriction enzyme panel was visualized on a 1% agarose gel post-stained with EtBr. The label 'P' represents supercoiled plasmid, while 'C' represents chromatinized plasmid. (Bottom right) The location of the restriction sites used are indicated on a diagram of the plasmid. (Bottom left) After quantification of the gel, the percent protection from cleavage experienced in the chromatinized plasmid was plotted versus the location of the cleavage sites on the top strand of the 601 sequence. Experiment 1 refers to the REAA experiment shown in the gel above, while experiment 2 refers to the REAA experiment without remodeler shown in *Figure 5F*. The grey shading indicates the borders of the 601 sequence, and the grey oval represents the corresponding nucleosome. (D) REAA experiment using the frequent cutter, HaeIII, to assess the remodeling activity around the chromatinized plasmid by the purified yChd1 chromatin remodeler. The resulting banding patterns were visualized on a 1.5% agarose gel post-stained with EtBr. Low molecular weight fragments indicate a high degree of HaeIII accessibility, while higher weight bands indicate protection from digestion. (E) Diagram showing the location of the restriction enzyme cleavage sites and the PAMs targeted by Cas9/sgRNA in the experiment shown in (F) and (G). (F) An accessibility assay was performed essentially as in C using either restriction enzymes or Cas9/sgRNAs in the presence or absence of the remodeler yChd1. The level of cleavage by the restriction enzyme panel (left) or Cas9/sgRNAs (right) was visualized on a 1.3% agarose gel post-stained with EtBr. A negative control was conducted with an sgRNA that had no sequence complementarity to the plasmid used (non-sense guide). The concentration of yChd1 used was the same as in panel (D) (G) Quantification of the gels shown in F. Percent protection from cleavage of the chromatinized plasmid in the presence or absence of the chromatin remodeler was calculated relative to the percent cleavage in the corresponding supercoiled plasmid control, and plotted at the location of the restriction enzyme cleavage sites or the center of the PAMs with respect to the 601 dyad.

## The chromatin remodeling enzyme yChd1 can restore access to nucleosomal DNA in vitro

The nucleosome landscape in eukaryotic chromatin is dictated by both intrinsic DNA sequence preferences as well as extrinsic factors such as chromatin remodeling enzymes (*Jiang and Pugh, 2009*; *Flaus and Owen-Hughes, 2011*; *Hargreaves and Crabtree, 2011*). In order to model how these dynamics affect Cas9 access to nucleosomal DNA in vitro, we turned to a chromatinized plasmid system. We dialyzed plasmid DNA containing a single 601 nucleosome positioning sequence with a sub-saturating quantity of purified histone octamers, and confirmed the quality of the resulting chromatin assemblies by MNase digestion (*Figures 5A,B*). We tested whether a nucleosome was positioned at the 601 sequence using restriction enzyme accessibility mapping, and found that sites within the 601 sequence were protected from digestion while sites immediately adjacent were not, suggesting precise positioning of a high occupancy nucleosome (*Figure 5C*). To recapitulate the effects of chromatin remodeling in vitro, we used a purified, truncated form of the Snf2-like chromatin remodeling enzyme Chd1 from *Saccharomyces cerevisiae* (yChd1), which had previously been shown to mediate nucleosome sliding in an ATP-dependent manner without additional co-factors (*Patel et al., 2011*; *2013*). To confirm yChd1 activity on our chromatinized plasmid, we used the frequent cutter, HaeIII, to digest the chromatin in the presence or absence of the remodeler. Upon addition of yChd1, we observed a shift toward lower molecular weight bands, indicative of dimished protection at HaeIII sites while still maintaining an overall chromatinized state (*Figure 5D*).

We next sought to test whether yChd1 could affect Cas9's ability to access nucleosomal DNA. Before addition of the remodeler to our chromatinized plasmid, we found that sites within the 601 nucleosome were strongly protected from cleavage by Cas9, consistent with our mononucleosome results (Figure 5E-G, PAM sites without remodeler). However, when the 601 nucleosome was remodeled by yChd1, as indicated by a loss of protection from restriction enzyme cleavage (approaching protection levels similar to those in the linker region), Cas9 cleavage efficiency was restored to around 80% of the corresponding naked plasmid control (*Figures 5E–G*). Notably, the percent protection at the EcoRI site adjacent to the positioned nucleosome did not decrease upon addition of yChd1, demonstrating that the decrease in protection along the 601 sequence was mediated by the nucleosome displacement activity of yChd1 rather than by a non-specific effect on cleavage efficiency. While our data with the chromatinized plasmid system confirm our findings that a well-positioned nucleosome provides a profound block to Cas9 cleavage, our further finding that chromatin

remodeling restores access to nucleosomal DNA provides one potential mechanism by which Cas9 may efficiently modify broad portions of eukaryotic genomes. This plasmid model could be further exploited to assay the activity of Cas9 at nucleosome-free and boundary sites, and thus derive bio-physical parameters governing Cas9-chromatin interactions.

## Discussion

Despite its swift success as a repurposed tool for gene editing, imaging, and transcription modulation, the ability of the prokaryotic CRISPR/Cas9 system to effectively navigate eukaryotic chromatin has remained poorly understood. Here, we show that the nucleosome, the basic unit of chromatin, poses a strong barrier to Cas9, both in vitro and in vivo. By masking ∼147 bp of DNA, the nucleosome effectively reduces the size of the eukaryotic genome available to Cas9. Previous studies using ChIP-seq to assay Cas9 binding have shown that off-target binding at PAM plus seed sequences more frequently occurs in regions of open chromatin (*Wu et al., 2014*; *O'Geen et al., 2015*). Our data expand upon these findings to show that the discrete pattern of nucleosome organization is able to modulate the efficiency of Cas9 binding and cleavage at on-target sites. The practical implications of these observations are underscored by our finding that accounting for nucleosome occupancy offers a significant improvement in predictive power for sgRNA design.

While our data show that nucleosomes strongly protect their DNA from Cas9 binding and cleavage in vitro, their organization in cells is not static. Transient displacement of nucleosomes occurs during replication, remodeling, and transcription. By adding the chromatin remodeling enzyme yChd1 to nucleosomes in vitro, we demonstrate that this displacement can in fact restore Cas9 access to DNA. However, despite brief exposure of nucleosomal DNA during remodeling and various other cellular processes, we still observe a clear anti-correlation between Cas9 activity and nucleosome occupancy in vivo, suggesting that the barrier to Cas9 target recognition exists even in a cellular environment. Indeed, the balance between nucleosome disruption, turnover, and repositioning in the cell leads to the average level of occupancy and positioning at each site observed by MNase-seq (*Valouev et al., 2011*; *Jiang and Pugh, 2009*). Thus, our data suggest that it is likely the overall effect of this average nucleosome positioning that leads to the observed anti-correlation with Cas9/sgRNA activity.

Additionally, it is important to note that there is likely a fundamental difference between applications that use dCas9 versus nuclease-active Cas9. Knock-down of transcription by CRISPRi likely requires persistent binding by dCas9 to continually block transcription, and would be largely ineffective during S-phase when transcription is globally shut down. In contrast, to make a genomic edit, Cas9 must succeed in cleaving DNA only once, and could potentially take advantage of nucleosome turnover during replication. The role these differences play in the dependence of Cas9 on nucleosome position is still not clear. We expect, however, that nucleosome position and occupancy will be of particular concern to applications that use the nuclease-dead Cas9 and require sustained binding. Future investigations into the role of cell cycle and nucleosome disruption may provide an additional piece to our understanding of the mechanism of Cas9 in eukaryotic cells. Furthermore, nucleosome organization represents only one aspect of eukaryotic chromatin, and thus, our results contribute a first step in understanding and exploiting how chromatin affects Cas9 activity to enable more sophisticated and precise rules for targeting Cas9.

## Materials and methods

### Analysis of CRISPRi sgRNA activity

The K562 dCas9-KRAB-BFP cell line was obtained from (*Gilbert et al., 2014*) and had been constructed from K562 cells obtained from ATCC. The resulting cell line tested negative for mycoplasma (MycoAlert Kit, Lonza, Basel, Switzerland) in regular screenings, and cytogenetic profiling by array comparative genomic hybridization (not shown) closely matched previous characterizations of the K562 cell line (*Naumann et al., 2001*). Data from 30 published (*Gilbert et al., 2014*) and unpublished screens (ML, BA, BB, CYP, MK, YC, JF, and JN, personal communication), conducted using the CRISPRi sgRNA library described in *Gilbert et al., (2014)* in K562 cells constitutively expressing dCas9-KRAB-BFP, were processed through a standardized pipeline adapted from *Bassik et al.*

*(2013)*, and *Kampmann et al. (2013)*. Briefly, sgRNA phenotypes were calculated as the $\log_2$ enrichment of sequencing read counts between two conditions (e.g. initial and final timepoints for growth screens, untreated and treated for drug/toxin screens) and normalized to cell doubling differences where appropriate. Most screens were conducted in duplicate, and sgRNA phenotypes from the duplicates were averaged. To determine hit genes, each gene was given an effect size (average of strongest 3 sgRNA phenotypes by absolute value) and a confidence value (Mann-Whitney p-value of all ~10 sgRNAs compared to negative controls), and hits were selected using a score that integrates effect size and statistical confidence (| effect Z score * $\log_{10}$ p-value | $\geqq$ 20 in any screen). For genes with multiple TSS, each TSS was analyzed separately and the gene was assigned the highest score. Finally, sgRNA phenotypes were extracted for hit genes from the screen in which the gene scored as a hit and normalized to the average of the strongest 3 phenotypes to generate the 'sgRNA activity score'.

sgRNA positions were defined as the genomic coordinate of the 3' G of the NGG PAM (all genomic coordinates referenced in this text are from hg19). TSS positions were determined from the FANTOM5 project annotation (Riken) (http://fantom.gsc.riken.jp/5/datafiles/phase1.3/extra/TSS_classifier/TSS_human.bed.gz; accessed March 2, 2015), using the downstream genomic coordinate of the corresponding 'p1@gene' BED file entry. All local averages were calculated in 50 bp windows centered around the indicated point. As sgRNA lengths including the PAM were ~24 bp and position was calculated relative to PAM, a window size of 50 bp captures all sgRNAs that directly neighbor the center point at either the 3' or 5' end. In order to quantify amplitude, the local averages were first smoothed using a low-pass Butterworth filter (N = 4, Wn = 0.03; SciPy signal processing module) and then peaks and troughs were calculated by determining the local maxima and minima, respectively. As described for *Figure 1D*, Peak 0 was defined as the local maximum closest to the TSS, peak 1 was defined as the next maximum downstream of peak 0, and troughs were defined as the minima immediately downstream of the respective peaks.

## MNase-seq data analysis
K562 MNase-seq data was obtained from the ENCODE consortium as processed continuous signal data (BigWig file format; accession number ENCFF000VNN, Michael Snyder lab, Stanford University). sgRNA target site signal was calculated as the average signal at all positions between the 5' end of the sgRNA and the 3' end of the PAM.

## RNA-seq data analysis
K562 RNA-seq data were obtained from the ENCODE consortium as transcript quantifications (accession number ENCFF485YKK, Thomas Gingeras lab, Cold Spring Harbor Laboratories). Genes were assigned expression levels in units of FPKM according to their highest-expressed transcript.

## Linear regression
Sequence score was calculated by passing the specified 30 bp target site for each sgRNA to the on_target_score_calculator.py script provided by Doench and colleagues (accessed October 9, 2015) (*Doench et al., 2014*). This sequence score, MNase signal as calculated above, and sgRNA length (base pairs in protospacer and PAM) were each compared to sgRNA activity scores by Pearson correlation. Linear fits of the specified parameters were computed using multidimensional linear regression (Sci-kit learn linear_model package), and correction of the activity scores for sequence and length features was performed by subtracting the predicted scores based on the combined fit.

## Ricin tiling screen
Screens for ricin susceptibility were performed essentially as previously described (*Gilbert et al., 2014*). Briefly, K562 cells constitutively expressing Cas9-BFP from an SFFV (spleen focus-forming virus) promoter were transduced with our previously described pooled sgRNA tiling library packaged into lentivirus for a multiplicity of infection below 1. Duplicate screens were infected and subsequently treated independently. Infected cells were allowed to recover for 2 days, then selected with 0.75 µg/mL puromycin (Tocris) for 2 days, and finally allowed to recover from puromycin treatment for 2 days. Cells were then cultured for 19 days and were either treated with three pulses of 0.5 ng/mL ricin administered for 24 hr and followed by re-suspension in fresh media, or passaged

untreated. Genomic DNA was harvested from the endpoint untreated and treated samples and processed for high-throughput Illumina sequencing as previously described (*Gilbert et al., 2014*). Screens were conducted at a minimum library coverage of 1000 cells per sgRNA, and sequenced to a median depth of ~500 reads per sgRNA. Phenotypes were calculated as $\log_2$ enrichments of read counts between untreated and treated conditions, normalized to cell doubling differences, and averaged between duplicates. Phenotype-signed Z scores were calculated by dividing all scores by the standard deviation of negative control phenotypes and then multiplying phenotypes by -1 for sgRNAs targeting genes shown to produce sensitizing phenotypes upon knockdown (*Bassik et al., 2013*) such that positive values represent 'expected' phenotypes.

## Protein purification

Mouse histones H2B(T115C) and H4 were recombinantly expressed in BL21(DE3)pLysS cells from expression plasmids gifted by Dr. Karolin Luger. Expression and purification of mH4 was conducted as previously described by the Luger lab (*Dyer et al., 2004*; *Luger et al., 1999*), while mH2B(T115C) was expressed and purified as described by the Cairns lab (*Wittmeyer et al., 2004*) with the following exception: after purification, histones were dialyzed against multiple changes of double distilled water and 1 mM β-mercaptoethanol (BME) before lyophilizing for storage. Purified recombinant mouse (*Mus Musculus*) histones H2A and H3 were gifts from Dr. Karolin Luger.

The labeling mutant, mH2B(T115C) was fluorescently labeled with ~five-fold molar excess Alexa Fluor 647 $C_2$ maleimide dye (Thermo Fisher Scientific, Waltham, MA, USA) as follows. 2 mg of lyophilized mH2B(T115C) was dissolved at 2 mg/mL in labeling buffer (7 M GuHCl, 50 mM Tris-HCl pH 7.5, 1 mM TCEP) and nutated at room temperature for 30 min. 1 mg of the dye was then dissolved in 50 µl of anhydrous DMF under Argon gas, and approximately half of the dye solution was slowly mixed with the dissolved mH2B(T115C) in the dark at room temperature to begin the labeling reaction. After nutating the reaction for 1 hr, the rest of the dissolved dye was slowly added, and the reaction was moved to 4°C overnight in the dark. In the morning, the reaction was quenched by adding over hundred-fold molar excess of BME.

Histone octamers were refolded and purified as previously described (*Dyer et al., 2004*). Specifically, 110 nmoles each mH3 and mH4 were refolded with 130 nmoles each mH2A and mH2B (T115C). The resulting octamer was concentrated using a 10,000 MWCO Spin-X UF concentrator (Corning, Tewksbury, MA), then purified on a Superdex 200 HR (10/30) column (GE Life Sciences, Pittsburgh, PA) using an Akta Explorer FPLC (GE Life Sciences, Pittsburgh, PA) at 0.2 mL/min. Selected fractions were concentrated, flash frozen, and stored at -80°C until use.

A new purification scheme was conceived to achieve exceptionally high purity (d/wt)Cas9 (*Figure 4—figure supplement 3*). Nuclease active *S. pyogenes* 6His-Cas9-HaloTag-NLS (HaloTag is a registered trademark of Promega, Madison, WI) was recombinantly expressed and purified from BL21(DE3)pLysS-Rosetta cells (Novagen/EMD Millipore, Darmstadt, Germany) using the expression plasmid pET302-6His-wtCas9-Halo-NLS, while the nuclease dead *S. pyogenes* 6His-dCas9(D10A, H840A)-HaloTag was recombinantly expressed and purified from BL21(DE3)pLysS cells using the expression plasmid pET302-6His-dCas9-Halo (*Knight et al., 2015*). Bacterial cultures were grown in Terrific Broth II (MP Biomedicals, Santa Ana, CA) at 37°C until an $OD_{600}$ reached 0.4. Cultures were then transferred to an ice bath for ~15 min until an $OD_{600}$ reached 0.5, at which point expression was induced with 0.2mM IPTG, and the cultures were moved to an 18°C shaker for 16 hr. Cells were harvested at 3000xg for 20 min, then resuspended in lysis buffer (500 mM NaCl, 50 mM Hepes pH 7.6, 5% glycerol, 1% Triton X-100, 10 mM imidazole, 1 mM benzamadine, 2.3 µg/mL aprotinin, 0.5 mM PMSF, and 1 tablet per 50 mL of Protease Inhibitor Cocktail (Roche, Basel, Switzerland). Cells were lysed using sonication on ice at 50% duty cycle, power 8, 30 seconds on, 1 minute off (Misonix/ Qsonica, LLC, Newtown, CT). The lysed cells were then ultracentrifuged at 4°, 40K rpms, for 40 min to remove cell debris. The supernatant was then allowed to bind to Ni-NTA agarose resin (Qiagen, Hilden, Germany) by nutating at 4°C for 30 min. The resin containing bound (d/wt)Cas9 was poured into a mini column (Bio-Rad, Hercules, CA) and washed with 10 column volumes (CV) of lysis buffer, and 5 CVs of 20 mM Imidazole buffer (same as lysis buffer but with 20 mM imidazole). Elution of the (d/wt)Cas9 was achieved using 250 mM Imidazole buffer (same as lysis buffer but with 250 mM imidazole), and fractions were checked on an SDS-PAGE gel. Chosen fractions were pooled and diluted to a starting NaCl concentration of 200 mM using Buffer A (0 M NaCl, 50 mM Hepes pH 7.6, 5% glycerol, 1 mM DTT, 0.5 mM PMSF). A 5 mL HiTrap Q-HP (GE Life Sciences, Pittsburgh, PA)

and a 5 mL HiTrap SP-HP (GE Life Sciences, Pittsburgh, PA) column were attached in tandem (Q first in line) and equilibrated on an Akta FPLC at 10% Buffer B (same as Buffer A except with 2 M NaCl). The pooled (d/wt)Cas9 was filtered, then loaded onto the tandem columns at 2 mL/min. The columns were washed with 10% Buffer B until $A_{280}$ and $A_{260}$ returned to baseline, at which point the Q column was removed and (d/wt)Cas9 was eluted from the SP column using a gradient from 10% to 50% Buffer B over 10 CVs. Fractions were chosen using SDS-PAGE, pooled, and dialyzed into storage buffer (200 mM NaCl, 50 mM Hepes pH 7.6, 5% glycerol, 1 mM DTT) (*Figure 4—figure supplement 3*). Aliquots were flash frozen in liquid nitrogen and stored at -80°C.

## Target DNA purification

To make the fluorescent DNA used in this study, the 601 DNA sequence was amplified by PCR from plasmid pBSIISK+601 (parent 601 sequence plasmid gifted by K Luger) with the primers listed below. Two different PCR products were produced; one labeling the Watson strand of the 601 sequence with a 5' Cy3 dye (IDT, Coralville, IA), and the other labeling the Crick strand. Large-scale (~2 mL) PCR reactions using in-house produced Pfu DNA polymerase were ethanol precipitated before loading onto a 20cm x 20cm 12% native TBE-PAGE gel for DNA purification. The PCR product was cut out of the gel, and the gel slice was crushed and soaked in 0.3 M sodium acetate pH 5.2 with multiple buffer changes. The pooled extract was ethanol precipitated, resuspended in 10 mM Tris-HCl pH 7.5, 10 mM NaCl, and stored at -20° until use.

### Watson primer pair
LW021: 5'- /5Cy3/ATCGGATGTATATATCTGACACGTGC -3'
LW022: 5'- ATCTGAGAATCCGGTGCCG -3'

### Crick primer pair
LW025: 5'- /5Cy3/ATCTGAGAATCCGGTGCCGAG -3'
LW030: 5'- ATCGGATGTATATATCTGACACGTGC -3'

## sgRNA production for in vitro experiments

sgRNA was produced by T7 transcription of a short DNA oligo template. To create this template, two oligos, one containing the T7 promoter and DNA target sequence, and the other encoding the invariable scaffolding of the sgRNA, were annealed and filled in using a single PCR cycle. The DNA template was ethanol precipitated, then transcribed using the T7 Quick High Yield RNA Synthesis Kit (New England Biolabs, Ipswich, MA) according to vendor instructions. The resulting sgRNAs were purified via 6% Urea-PAGE gel. The correct band was cut out and crushed and soaked in 0.3M sodium acetate pH 5.2. After ethanol precipitation, and resuspension in RNase-free water, aliquots were stored at -80°C until use. SgRNAs targeting the 601 sequence included 20 bp of complimentarity 5' to the targeted PAM, with the exception of PAM 4, which has room for only 19 bp of complimentarity. SgRNAs were used against PAMs 1, 3–15, and 17–20. PAM 2 did not have a long enough target sequence available within the 601. The non-sense guides used in this study contained the target sequences 5'-ACATGTTGATTTCCTGAAA-3' or 5'-GATTTCACCTCTCAGCGCAT-3'.

### Template oligonucleotides
5'-TTAATACGACTCACTATAGNNNNNNNNNNNNNNNNNNNNGTTTTAGAGCTAGAAATAGC -3'
5'-AAAAAGCACCGACTCGGTGCCACTTTTTCAAGTTGATAACGGACTAGCCTTATTTTAACTTGCTATTTCTAGCTCTAAAAC -3'

## Nucleosome assembly

Nucleosomes were assembled by salt gradient dialysis as previously described (*Dyer et al., 2004*) using 50 µL home-made dialysis buttons and ~1 µM DNA. The Watson and the Crick labeled 601 DNA were assembled into separate nucleosomes. The best DNA:octamer molar ratio for optimal nucleosome assembly was selected by titrating octamer. The most homogenous assembly as judged by 6% Native TBE-PAGE gel was chosen for future assays.

## Cleavage and binding assays

Cleavage and binding assays in *Figures 3,4* and their corresponding supplements were both conducted in the same manner, exchanging dCas9 for wtCas9 during binding reactions. First, the complete ribonucleoprotein complex was formed by incubating five-fold molar excess of the chosen sgRNA with (d/wt)Cas9 at 37°C for 10 min. Next, the DNA substrate (either naked DNA or nucleosome) was added to 40 nM (five-fold less than Cas9), and the reactions were returned to 37° for an hour. Reaction buffer contained 20 mM Hepes pH 7.5, 100 mM NaCl, 5 mM $MgCl_2$, 1 mM EDTA, and 2.5 mg/mL insulin (Roche, Basel, Switzerland). Importantly, we found that including a nonspecific protein such as insulin prevents the nucleosomes from falling apart and getting lost to surfaces during binding and cleavage assays (*Figure 4—figure supplement 1B*). Additionally, inclusion of insulin reduces nonspecific protein-protein interactions and aggregation. Binding reactions also contained 14.6% sucrose to allow direct loading onto a native gel. At 1 hr, binding reactions were loaded onto a pre-run 6% native TBE-PAGE gel at 4°C, while cleavage reactions were stopped using a 5X stop buffer (250 mM EDTA, 2% SDS), and prepared to load onto a 10% Urea-PAGE gel by adding 2X loading buffer (95% formamide, 20% DMSO, 5 mM EDTA, 0.025% Orange G) and heating to 95°C for 5 min before snap cooling on ice. Binding and cleavage reactions were repeated at least twice each for most of the PAMs targeted. Gels were imaged using a PharosFX Plus (Bio-Rad, Hercules, CA) and quantified using Image Lab (Bio-Rad, Hercules, CA). For binding and cleavage reactions, gels were scanned in the Cy3 fluorescence channel, while Alexa Fluor 647 fluorescence was also imaged for binding reactions. For cleavage reactions, labeling of either the Watson or Crick strand was chosen such that the fluorophore was always attached to the strand complimentary to the sgRNA used.

## Structure alignment and solvent accessibility

Molecular graphics and analyses were produced using the UCSF Chimera package (supported by NIGMS P41-GM103311) (*Pettersen et al., 2004*). The crystal structures for the 601 nucleosome (*Vasudevan et al., 2010*, PDBID 3LZ0) and the Cas9-sgRNA-DNA ternary complex (*Anders et al., 2014*, PDBID 4UN3) were superimposed by aligning the DNA path in both structures using MatchMaker and manual manipulation. The solvent accessible surface area of the DNA in the 601 nucleosome structure was computed using residue areaSAS.

## Reconstitution of nucleosomes on a plasmid (chromatin assemblies)

Supercoiled plasmid was produced using a Qiagen Maxi prep kit (Hilden, Germany). Histone octamers were prepared from purified histones as described above. DNA and histone octamer were mixed at a weight ratio of 1:0.8 (DNA to octamer) at a concentration of 0.4 µg/µL of DNA in 1X TE with 2 M NaCl. 50 µL homemade dialysis buttons were used to dialyse the solution by a step-wise gradient in 500 mL at room temperature. The gradient was as follows; 1.5 hr in 1 M NaCl in TE, 2 hr in 0.8 M NaCl in TE, 1.5 hr in 0.6 M NaCl in TE, and 2 hr in 0.05 M NaCl in TE. The resulting chromatin was stored at 4°C until use.

## Micrococcal nuclease digestion assays of chromatin assembly reactions

MNase assays were performed essentially as in *Torigoe et al. (2013)* and *Alexiadis (2002)*. MNase (Sigma-Aldrich, St. Louis) was resuspended at 200 units/mL in water and then disluted at 1:100, 1:500, or 1:1000 in a solution of 1X MNase reaction buffer (50 mM Tris-HCl pH7.9 and 5 mM $CaCl_2$ dihydrate), 0.1 mg/mL insulin, and 10% glycerol. Chromatinized plasmid assemblies in 1X MNase buffer were added to each MNase dilution. As a control, an equal amount of unchromatinized super-coiled plasmid was added to the lowest MNase dilution. Each reaction was incubated for 11 min at room temperature, then stopped in a solution with final concentrations of 20 mM EGTA, 200 mM NaCl, 1% SDS, 20 mg/mL GlycoBlue (Thermo Fisher Scientific, Waltham, MA), and 13.4 mg/mL Proteinase K (Roche, Basel, Switzerland) and incubated at 37°C for 30 min. Reactions wer Phenol/Chloroform/Isoamyl Alcohol extracted, ethanol precipitated, and run on a 1.3% agarose gel in 0.5X TBE. 3 µg of the 123 bp DNA ladder (Thermo Fisher Scientific, Waltham, MA) was used as a size standard.

## Restriction enzyme accessibility assays

Either chromatin assemblies or super-coiled plasmid with equivalent amounts of DNA were added to restriction enzyme master mixes for final concentrations of 1X CutSmart buffer (New England Biolabs, Ipswich, MA), 6 ng/μL DNA, and 0.4 U/μL of the indicated restriction enzyme (New England Biolabs). Reactions were incubated for 1 hr at NEB recommended temperatures. Reactions were stopped in final concentrations of 15 mM EDTA, 0.75% SDS, 150 mM NaCl, and 15 mg/mL Proteinase K and incubated at 37°C for 30 min. DNA was extracted with Phenol:Chloroform:Isoamyl Alcohol and ethanol precipitated, then resuspended in 1X CutSmart buffer and linearized by digesting with 1 U/μL DraIII-HF (New England Biolabs) at 37°C for 1 hr. The full reactions were run on an agarose gel and quantified as above.

## Chromatin remodeling assays

Chromatin remodeling assays were performed simmilar to *Torigoe et al. (2013)* and *Alexiadis (2002)*. Reactions contained 40 mM NaCl, 0.1 mg/mL BSA, 25 mM Tris-Acetate pH 7.5, 10 mM Mg-Acetate, 1 mM DTT, 1.2 mM ATP, and 6 ng/μL chromatinized or supercoiled plasmid. For restriction enzyme accessibility, the indicated restriction enzyme was added at 0.5 U/μL. For Cas9 accessiblity, Cas9/sgRNA ribonucleoproteins assembled as described above were added at 15.62 nM. A truncated version of the chromatin remodeling enzyme yChd1 was used spanning amino acids 118–1274, also referred to as $yChd1_{\Delta NC}$ (gift of Dr. Ashok Patel and Dr. Gregory Bowman), and was added to the indicated reactions at a 0.2 μM final concentration. Reactions were incubated at 27°C for 1 hr, then processed as with the restriction enzyme accessibility assay above.

# Acknowledgements

We would like to thank Dr. Karolin Luger, Pamela Dyer, and Dr. Uma Muthurajan for reagents and training in preparing and working with nucleosomes, and Dr. Ashok Patel and Dr. Gregory Bowman for the kind gift of yChd1. We would also like to thank Dr. Manuel Leonetti, Dr. Britt Adamson, Ben Barsi-Rhyne, Dr. Chong Y Park, Dr. Martin Kampmann, Yuwen Chen, Dr. Jonathan Friedman, and Dr. Jodi Nunnari for generously sharing unpublished screening data for determination of sgRNA activity. Additionally, we would like to thank members of the Tjian and Weissman labs, in particular, Dr. Elisa Zhang, Dr. Liangqi Xie, and Chiahao Tsui for their help with in vitro reagents, and Dr. Alex Fields and Joshua Dunn, for helpful discussions and assistance.

# Additional information

## Competing interests

MAH, LAG and JSW: Filed a patent application related to CRISPRi screening techonology. RT: President of the Howard Hughes Medical Institute (2009-present), one of the three founding funders of eLife, and a member of eLife's Board of Directors. The other authors declare that no competing interests exist.

## Funding

| Funder | Grant reference number | Author |
| --- | --- | --- |
| Howard Hughes Medical Institute | | Max A Horlbeck<br>Lea B Witkowsky<br>Benjamin Guglielmi<br>Joseph M Replogle<br>Luke A Gilbert<br>Jacqueline E Villalta<br>Robert Tjian<br>Jonathan S Weissman |
| National Institutes of Health | P50 GM102706 | Max A Horlbeck<br>Joseph M Replogle<br>Luke A Gilbert<br>Jacqueline E Villalta<br>Jonathan S Weissman |

| | | |
|---|---|---|
| UCSF Medical Scientist Training Program | | Max A Horlbeck<br>Joseph M Replogle |
| National Institutes of Health | U01 CA168370 | Max A Horlbeck<br>Joseph M Replogle<br>Luke A Gilbert<br>Jacqueline E Villalta<br>Jonathan S Weissman |
| National Institutes of Health | R01 DA036858 | Max A Horlbeck<br>Joseph M Replogle<br>Luke A Gilbert<br>Jacqueline E Villalta<br>Jonathan S Weissman |
| California Institute for Regenerative Medicine | RB4-06016 | Lea B Witkowsky<br>Benjamin Guglielmi<br>Sharon E Torigoe<br>Robert Tjian |
| Leukemia and Lymphoma Society | | Luke A Gilbert |

The funders had no role in study design, data collection and interpretation, or the decision to submit the work for publication.

**Author contributions**

MAH, Conceived of and conducted the in vivo data analysis and experiments, Helped write this report; LBW, Conceived of and conducted the in vitro experiments and analysis, Helped write this report; BG, Contributed to conception and interpretation of the in vitro experiments, Edited this report; JMR, Contributed analysis of in vivo tiling screen data; LAG, JEV, Conducted tiling screens; SET, Contributed to chromatin remodeling experiments; RT, Supervised the in vitro experiments carried out by LBW, Helped write this report; JSW, Supervised the in vivo experiments carried out by MAH, Helped write this report

# Additional files

## Supplementary files

• Supplementary file 1. Table S1. CRISPRi sgRNA annotations, activity scores, and target site MNase signal, related to *Figure 1*.

• Supplementary file 2. Table S2. Ricin tiling library sgRNA annotations, phenotype scores, and target site MNase signal, related to *Figure 2*.

## Major datasets

The following previously published datasets were used:

| Author(s) | Year | Dataset title | Dataset URL | Database, license, and accessibility information |
|---|---|---|---|---|
| FANTOM Consortium | 2014 | A promoter-level mammalian expression atlas | http://fantom.gsc.riken.jp/5/datafiles/phase1.3/extra/TSS_classifier/TSS_human.bed.gz | Publicly available at Functional Annotation of the Mammalian Genome (FANTOM) |
| ENCODE Consortium/Snyder | 2011 | Determinants of nucleosome organization in primary human cells. | https://www.encodeproject.org/experiments/ENCSR000CXQ/ | Publicly available at Encode: Encyclopedia of DNA Elements (accession number: ENCFF000VNN) |

| ENCODE Consortium/Gingeras | 2014 | Landscape of transcription in human cells. | https://www.encodeproject.org/experiments/ENCSR000AEL/ | Publicly available at Encode: Encyclopedia of DNA Elements (accession number: ENCFF485YKK) |
|---|---|---|---|---|

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
