## [Decision Letter]

Thank you for submitting your work entitled "Nucleosomes Impede Cas9 Access to DNA in vivo and in vitro" for consideration by *eLife*. Your article has been favorably evaluated by Jessica Tyler (Senior editor) and two reviewers, one of whom is a member of our Board of Reviewing Editors.

The reviewers have discussed the reviews with one another and the Reviewing Editor has drafted this decision to help you prepare a revised submission.

Summary:

This manuscript by Horlbeck, Witkowsky et al. deals with the effects of nucleosomes within eukaryotic genomes on the binding and activity of prokaryotic Cas9 proteins. Given the widespread usage of Cas9 and catalytically inactive dCas9 proteins to modulate metazoan gene expression, it is very important to understand how histone occupancy impinges on targeting of the Cas9 module. Previous studies have suggested that nucleosomes may block accessibility of sgRNAs and Cas9, but these involved limited numbers of sgRNAs and resulted in only in the generalization that Cas9 seems to best target regions that are, for example, DNaseI accessible.

This work tackles the topic head on, analyzing thousands of sgRNAs and MNase data in vivo, coupled with careful biochemical assays. The authors show unequivocally that the organization of DNA in a nucleosome impedes access of Cas9, irrespective of the orientation of the PAMs. The in vitro experiments are technically sound and support the conclusions. Additional strengths of this work include: 1) the very nice anti-correlation between nucleosome occupancy and sgRNA activity shown in Figure 1; 2) Figure 1, where increased nucleosome phasing is shown to increase the positional dependence on activity of sgRNAs.

This finding is important, the data is clean and the manuscript well written.

However, the work does have several weaknesses that should be addressed prior to publication, as noted below.

Essential revisions:

1) The data suggesting that nucleosomes completely block both binding and cleavage would be extremely interesting if it were better documented. In particular, the authors should investigate how longer DNA sequences behave when assembled into nucleosomes, or to test short nucleosomal arrays.

For example, can the system 'invade' the nucleosome when the PAM is located right outside of the edge of the 147 bp that are protected?

As the data stands, Figure 3 and Figure 4 lack a good positive control. It is important to show that a region of DNA outside the nucleosome can be bound and cleaved by Cas9 in this assay, to confirm that things are working as anticipated.

We recommend that the authors perform a few experiments with a longer DNA fragment containing the Widom 601 element (i.e. ~200 bp or so) or other sequence, and target a non-nucleosomal portion of this DNA with a sgRNA-Cas9.

Additionally, the 601 sequence is a bit atypical in its stability and 'rigidity', and this should be acknowledged.

2) Moreover, based on what is shown in the current manuscript, I am not sure one can conclude that 'breathing' on the edges of the nucleosome has no effect on sgRNA-Cas9 binding or activity. There are no sgRNAs targeting the far edge of the nucleosome, and the 2-3 that target the near edge are very poor at binding and cleaving naked DNA (PAM 1, 3, 4). If one were to calculate the 'effect of nucleosomes' as the ratio of binding or cleavage on naked DNA/ nucleosomal DNA, the PAM sequences on the edge would show smaller differences than those in the middle, which is not what the authors conclude. If the authors can't find a way to further probe the DNA farther away from the dyad, they should probably be cautious about what they conclude from this data.

3) In Figure 1 and Figure 2 the difference in sgRNA activity score is only significant for MNase signals between 0 and 0.75, with little or the opposite trend appearing as nucleosome occupancy becomes higher than 0.75. This raises a number of questions and makes me wish the authors had looked more closely at the sgRNAs that fall in regions with MNase signal between 0 and 0.75.

Is a trend apparent within the range of 0 and 0.75? This finding would be much more compelling if one could see a trend of decreasing activity across several intervals of MNase signal.

Or does the data indicate that a region needs an MNase signal near zero to be targeted well? This would important to note if true.

What are the numbers of sgRNAs in each bin? The figure or legend would benefit from a better designation of what N= for each group.

4) Figure 1 makes the point that thinking about nucleosome occupancy improves one's ability to predict the effectiveness of a given sgRNA. However, the Pearson R2 values reported here are still very low, indicating that even when considering sequence, length and MNase signal, one still has very little idea of whether an sgRNA will work. Can the authors comment on why they think our predictive power is still so poor? Are there other components (histone modifications nearby, gene activity level) that the authors might comment on, or investigate that could also be of use in defining a 'good' sgRNA?

---

## [Author Response]

*Essential revisions: 1) The data suggesting that nucleosomes completely block both binding and cleavage would be extremely interesting if it were better documented. In particular, the authors should investigate how longer DNA sequences behave when assembled into nucleosomes, or to test short nucleosomal arrays.*

*For example, can the system 'invade' the nucleosome when the PAM is located right outside of the edge of the 147 bp that are protected?*

Our intent in conducting the in vitro experiments was to determine whether nucleosomes were sufficient to block the access of Cas9 to DNA and thereby explain the in vivo results of Figure 1 and Figure 2. While identifying the precise boundary between protected and accessible DNA is an interesting question, we feel this is outside the scope of this project, and will depend strongly on the sequence and environment of the nucleosome in question. In addition, the in vivo measurement of nucleosome position through MNase-seq captures the average nucleosome occupancy at a given site rather than reporting on fixed mononucleosomes, and this average occupancy is likely to explain our observation of differential Cas9 accessibility in vivo. We have revised the Discussion to clarify this point (second paragraph).

As the data stands, Figure 3 and Figure 4 lack a good positive control. It is important to show that a region of DNA outside the nucleosome can be bound and cleaved by Cas9 in this assay, to confirm that things are working as anticipated.

Because the 601 sequence used in Figure 3 and Figure 4 assembles a minimal mono-nucleosome, there is no DNA outside the positioning sequence available to target. The naked DNA experiments provide some control for this concern as they show that indeed, Cas9 is working as anticipated, as it, controls for the sequence dependence of Cas9 cleavage. However, we believe that our new data provides a more complete control to address the reviewers’ concern. We show that in the context of a chromatinized plasmid (Figure 5), we observe near-absolute protection of DNA along the 601 sequence, fully consistent with our previous in vitro results using a minimal mono-nucleosome. To modulate the accessibility of the DNA within the 601 nucleosome, we introduce a remodeler, yChd1. We observe that chromatin remodeling restores the accessibility of DNA to Cas9 at the same underlying sequences previously observed to be fully protected from cleavage. This result shows that the observed protection is due to nucleosome positioning and not another aspect of the experimental set-up (e.g., the buffer conditions).

*We recommend that the authors perform a few experiments with a longer DNA fragment containing the Widom 601 element (i.e. ~200 bp or so) or other sequence, and target a non-nucleosomal portion of this DNA with a sgRNA-Cas9.*

As above, we feel this concern can be addressed by our chromatinized plasmid and remodeling experiment but we now indicate in the text that this would be a valuable future direction (Results, last sentence).

*Additionally, the 601 sequence is a bit atypical in its stability and 'rigidity', and this should be acknowledged.*

We appreciate this point and now address it in the text (subsection “Nucleosomes are sufficient to fully block cleavage by Cas9 in vitro”, second paragraph).

*2) Moreover, based on what is shown in the current manuscript, I am not sure one can conclude that 'breathing' on the edges of the nucleosome has no effect on sgRNA-Cas9 binding or activity. There are no sgRNAs targeting the far edge of the nucleosome, and the 2-3 that target the near edge are very poor at binding and cleaving naked DNA (PAM 1, 3, 4). If one were to calculate the 'effect of nucleosomes' as the ratio of binding or cleavage on naked DNA/ nucleosomal DNA, the PAM sequences on the edge would show smaller differences than those in the middle, which is not what the authors conclude. If the authors can't find a way to further probe the DNA farther away from the dyad, they should probably be cautious about what they conclude from this data.*

This point is well taken. In targeting sites near the edge of the nucleosome, we were limited by the availability of PAMs with a corresponding 20bp target site in the 601 sequence. We have accordingly adjusted our conclusions in the manuscript (subsection “Nucleosomes are sufficient to fully block cleavage by Cas9 in vitro”, second paragraph). Nevertheless, we do express the ratiometric effect of nucleosomes as “% Protection” in Figure 4 (now revised for clarity), and observe that all sgRNAs show greater than 95% protection regardless of their position relative to the dyad.

3) *In Figure 1 and Figure 2 the difference in sgRNA activity score is only significant for MNase signals between 0 and 0.75, with little or the opposite trend appearing as nucleosome occupancy becomes higher than 0.75. This raises a number of questions and makes me wish the authors had looked more closely at the sgRNAs that fall in regions with MNase signal between 0 and 0.75.*

*Is a trend apparent within the range of 0 and 0.75? This finding would be much more compelling if one could see a trend of decreasing activity across several intervals of MNase signal.*

*Or does the data indicate that a region needs an MNase signal near zero to be targeted well? This would important to note if true.*

*What are the numbers of sgRNAs in each bin? The figure or legend would benefit from a better designation of what N= for each group.*

We thank the reviewers for these suggestions. We have adjusted the boxplot bins in Figure 1, Figure 2, and Figure 2—figure supplement 1 to better demonstrate the relationship between MNase signal and sgRNA activity. In all three cases, we do in fact observe a decreasing trend in activity between the 0 and 0.8 bins. As before, this decrease is most pronounced in the upper quartiles of the sgRNA activity distribution, consistent with our hypothesis that low nucleosome occupancy enables efficient targeting by CRISPR but is not the sole factor in determining efficiency. We have also added N values for each bin directly to the plot, along with Mann-Whitney p-values comparing each bin to the lowest MNase signal bin.

4) *Figure 1 makes the point that thinking about nucleosome occupancy improves one's ability to predict the effectiveness of a given sgRNA. However, the Pearson R2 values reported here are still very low, indicating that even when considering sequence, length and MNase signal, one still has very little idea of whether an sgRNA will work. Can the authors comment on why they think our predictive power is still so poor? Are there other components (histone modifications nearby, gene activity level) that the authors might comment on, or investigate that could also be of use in defining a 'good' sgRNA?*

In conducting the analysis displayed in Figure 1 and Figure 2, we used simple linear regression to test whether the observed relationship between nucleosome positioning and sgRNA activity was orthogonal to sequence features, and indeed found that each of these provided modest predictive power that was independent of the other variables. As these results are derived from screens using our CRISPRi v1 library, all sgRNAs are already pre-selected for several features – most notably distance to the TSS – which are strongly predictive of activity and in effect bias towards nucleosome free regions.

We have since used the CRISPRi sgRNA activity data generated for this study as training data for a comprehensive machine learning approach. The resulting sgRNA scoring model achieved a Pearson R2 of ~0.32, sufficient to call highly active sgRNAs with a Receiver Operating Characteristic area under the curve (ROC-AUC) value of 0.81. Again, this result is super-imposed on our v1 library design rules; application of the scoring model to our ricin gene tiling dataset, which was not subjected to the v1 rules, achieved an ROC-AUC of 0.91. This approach differs from the linear regression shown here by accounting for non-linear relationships (e.g. MNase values as displayed in Figure 1 are not strictly linear) and by incorporating several other features (e.g. higher order sequence features, sgRNA secondary structure). However, we believe the development, characterization, and validation of this sgRNA activity model which led to the construction and testing of a v2 CRISPRi library, is beyond the scope of this study but have added references to this work in the revised manuscript (subsection “CRISPRi activity is periodic and out-of-phase with nucleosome positioning”, last paragraph).